# Boosting Training-Free Composed Image Retrieval with Tools

## Abstract

Composed Image Retrieval (CIR) retrieves a target image that preserves the reference image's content while applying user–specified textual edits. Training-free zero-shot CIR (ZS-CIR) has progressed by casting the task as text-to-image retrieval with pretrained vision–language models, prompting multimodal LLMs to produce target captions. However, these approaches are hindered by frozen priors and a mismatch between free-form text and the retriever's embedding space. In this work, we introduce TaCIR, a training-free, tool-augmented agent for ZS-CIR that jointly reasons over the reference image and manipulation text, optionally consults external tools, and instantiates the inferred edit as a visual proxy. This proxy grounds implicit intent and reduces text–based retrieval misalignment by enabling also image–to-image image comparisons in the retriever. A single, tool-aware, chain-of-thought prompt emits both an initial target description and an executable tool call; when a tool is invoked, the synthesized evidence is fed back to refine the description and guide retrieval. TaCIR requires no task-specific training and remains inference-efficient. Across four benchmarks and three CLIP backbones, TaCIR yields consistent improvements over strong training-free baselines, with average gains of 2.20% to 4.16%, establishing a new state of the art for training-free ZS-CIR while providing interpretable intermediate visualizations.

## 1 Introduction

Composed Image Retrieval (CIR) Vo et al. (2019) aims to retrieve a target image that remains visually similar to a reference image while incorporating modifications specified by user-provided manipulation text. Unlike traditional image retrieval Datta et al. (2008), which relies solely on unimodal features, CIR leverages both visual and textual cues to better capture the user intent. This multimodal formulation enables users to specify desired changes to reference images, clarifying intent and improving retrieval accuracy. Consequently, CIR has attracted increasing interest in internet search and e-commerce Chen et al. (2020); Saito et al. (2023), where it supports tasks such as scene image search with object manipulation and product recommendations with attribute modification.

To avoid costly annotation procedure and potential generalization issues caused by training, zero-shot CIR is emerging as the leading paradigm for CIR Saito et al. (2023); Baldrati et al. (2023); Tang et al. (2024). Recent approaches Karthik et al. (2024); Tang et al. (2025b) in this setting exploit the representation capabilities of pretrained multimodal large language models (MLLMs) and contrastive vision language models (VLMs) to convert CIR into a text-to-image retrieval problem. Specifically, they use an MLLM (*e.g.,* GPT-4o OpenAI (2022), Qwen2.5-VL Wang et al. (2024)) to produce an explicit description of the target image from the reference image and the manipulation text. This target description is then used for performing text-to-image retrieval within the VLMs (*e.g.,* CLIP Radford et al. (2021)) shared semantic space. This line of approaches is not only effective, but allows to tackle CIR without training, with the costs and biases derived from the latter.

While effective, training-free methods for CIR rely on two key assumptions. The first is that MLLM's domain knowledge and priors suffice to model the user intent. This implies that MLLMs can fully capture detailed information within the user query, such as fine-grained attributes and compatibility rules. This is often not true in practice, as MLLMs struggle with compositional understanding Ma et al. (2023); Li et al. (2024a); Kil et al. (2024); Mitra et al. (2024). The second is that the target caption produced by the MLLM can be easily processed by the retriever, *i.e.,* CLIP. This

assumption is also brittle as MLLMs' output might be verbose, prompt-sensitive, and not calibrated to the retriever's text encoder capabilities.

To address these challenges, we introduce a **T**ool-augmented **a**gent for training-free **C**omposed **I**mage **R**etrieval (**TaCIR**). To overcome the potential limited domain knowledge, TaCIR enables MLLMs to go beyond their frozen priors by consulting external resources. Specifically, the agent has access to tools (*i.e.*, web search, generative models) that can instantiate visual examples to more precisely infer the user intent for ambiguous queries. The presence of visual examples also allows us to overcome potential misalignment between MLLMs and retrievers. In fact, these examples act as explicit visual proxies for the target and can be used to compute image-to-image retrieval scores. By integrating extra-model knowledge and rendering user intent in pixels, TaCIR improves both the faithfulness of retrieval and interpretability through intermediate visualizations.

**Contributions.** To summarize: (1) We propose a training-free agent for CIR that jointly processes the reference image and manipulation text, acquires specialized extra-model knowledge when needed, and converts the inferred target edits into a synthesized visual proxy; (2) we use the visual proxy and target description to make the implicit manipulation cues explicit, allowing for both text-to-image and image-to-image matching, thereby reducing text–retrieval mismatch.(3) On four CIR benchmarks, TaCIR achieves consistent gains over MLLM-based training-free methods while maintaining inference efficiency, establishing a new state of the art for zero-shot CIR.

## 2 RELATED WORKS

**Composed Image Retrieval** (CIR) retrieves an image that reflect textual edits to a reference one Vo et al. (2019). While supervised methods have been proposed to tackle this task Liu et al. (2021); Baldrati et al. (2022), they rely on annotated triplets (*i.e.,* reference image, manipulation text, target image) to train task-specific models performing late fusion. Zero-Shot CIR Saito et al. (2023); Baldrati et al. (2023); Tang et al. (2024)has emerged as a solution to sidestep the annotation cost, with early approaches using textual inversion Baldrati et al. (2023); Tang et al. (2024) to convert images into text for later CLIP-based retrieval. However, this image-to-text mapping may miss fine-grained attributes essential for the task. More recent diffusion-based variants Gu et al. (2023); Wang et al. (2025); Li et al. (2025) address this by generating a visual proxy for every query, but consequently incurring in high computational cost. Differently, training free approaches use (M)LLMs to infer edits from the reference and text (e.g., CIReVL Karthik et al. (2024), OSrCIR Tang et al. (2025b)). Despite these progresses, these models can still miss important target details or yield captions misaligned with the retriever. We adopt a training-free, tool-augmented agent that uses an MLLM to choose which tool to call and whether to generate a visual proxy, producing proxies only when needed and allowing multi-step use of different tools across iterations. This flexible design remains efficient while improving faithfulness and retrieval accuracy compared with ensembles and diffusion-based methods (*e.g.,* Li et al. (2025); Gu et al. (2023)).

**Vision and Language Pre-training Models.** Vision–language pretraining (VLP) models such as CLIP Radford et al. (2021) align images and text from large image–text corpora, enabling broad zero-shot transfer Zhou et al. (2022); Song et al. (2022); Li et al. (2022); Alayrac et al. (2022); Li et al. (2023); Shi et al. (2023; 2024); Hummel et al. (2024). Multimodal LLMs, including LLaVA Liu et al. (2023) and GPT-4 family OpenAI (2024a;b), integrate visual inputs within LLM architectures and offer stronger multimodal reasoning. Retrieval-oriented variants (e.g., Com-CLIP Jiang et al. (2024); Chen et al. (2023); Li et al. (2024b); Sun et al. (2021)) further improve cross-modal matching. Recent training-free CIR shows that an MLLM coupled with a retriever can already be effective (e.g., CIReVL Karthik et al. (2024), OSrCIR Tang et al. (2025b)), while performance is bounded by frozen model knowledge and caption–retriever mismatch. We instead use an augmented MLLM that issues targeted queries to external resources to ground intent and create visual proxies, enabling effective CIR without additional training.

**Tool-based Agent for LLMs and MLLMs.** Recent studies highlight that relying solely on the parametric knowledge of (M)LLMs and VLMs is insufficient for complex multimodal tasks, motivating a shift toward tool-augmented reasoning and retrieval. For instance, AVIS Hu et al. leverages LLMs as planners that dynamically call web and image search for knowledge-intensive VQA; Dyn-VQA/OmniSearch Li et al. and mR$^2$AG Zhang et al. (a) integrate retrieval and reflection to mitigate hallucinations; and Vision Search Assistant Zhang et al. (b) explicitly frames MLLMs as

multimodal search engines. Similar ideas extend to agentic tasks(*i.e.,* SeeAct Zheng et al. and VisualWebArena Koh et al.), which showcase iterative tool use in open web environments. From a training perspective, T3-Agent (a.k.a. "MMAT") Gao et al. further enhances MLLMs' tool-selection ability through trajectory tuning. Collectively, these works establish retrieval-augmented tool use as a paradigm for advancing MLLMs reasoning. Building on this line, our work introduces a specific tool-agumented reasoning pipeline into composed image retrieval, enabling more accurate and robust multimodal retrieval under compositional queries.

# 3 METHODOLOGY

## 3.1 PRELIMINARIES

Given a reference image $I_r \in \mathcal{I}$, and a manipulation text $T_m$ in the textual space $\mathcal{T}$, that specifies hypothetical semantic changes to the reference, zero-shot CIR (ZS-CIR) aims to retrieve images from a database $\mathcal{D}$ that are visually similar to $I_r$ while also reflecting the modifications described in $T_m$. To achieve this without training, such methods employ a pretrained VLM (*e.g.* CLIP) as a retriever. The VLM is composed of an image encoder $\Psi_I : \mathcal{I} \to \mathcal{Z}$ and a text encoder $\Psi_T : \mathcal{T} \to \mathcal{Z}$ mapping images and text, respectively, into the shared space $d$-dimensional $\mathcal{Z} \in \mathbb{R}^d$. Moreover, they assume the presence of an MLLM $\Psi_M$ mapping multimodal inputs into textual output.

To perform zero-shot CIR in a training-free manner, standard approaches (*e.g.,* Karthik et al. (2024); Tang et al. (2025b) directly generate a target image description from the reference image and manipulation text. Specifically, let us denote with $F : \mathcal{I} \times \mathcal{T} \to \mathcal{T}$ a generic function that produces the target image description $T_t$ from the query image and modification, *i.e.,*, $T_t = F(I_r, T_m)$. In practice, $F$ is usually instantiated via $\Psi_M$. Standard text-to-image retrieval then scores each candidate image in $\mathcal{D}$ using cosine similarity with $T_t$ in the shared representation space, returning in output the image with the maximum similarity, *i.e.,*:

$$I_t = \underset{I \in \mathcal{D}}{\arg\max} \frac{\Psi_I(I)^\top \Psi_T(T_t)}{\|\Psi_I(I)\| \, \|\Psi_T(T_t)\|}. \tag{1}$$

Performing CIR via Eq. equation 1 assumes that the function $F$ has full domain knowledge and can easily capture the user intent. However, directly generating $T_t$ from $(I_r, T_m)$ with $F$ (and its constituent frozen MLLM) can be challenging as (i) under-specified or *implicit* cues and domain-specific constraints may not be resolved by language alone; (ii) the generated target image description can be verbose or poorly calibrated for the text encoder of the retriever $\Psi_T$, as the MLLM has no prior on the input expected by the latter. To address this, we propose an adaptive framework that allows the MLLM to *optionally* consult external tools to enhance reasoning. As shown in Figure 1, the MLLM first processes the reference image $I_r$ and manipulation text $T_m$ and decides whether to invoke an external tool to obtain a **tool-generated image** that serves as a visual proxy. The final target description $T_t$ is then obtained by a refinement step from the original inputs together with this proxy. We name our approach TaCIR. In the following, we describe the component of our framework.

## 3.2 TOOL-AUGMENTED AGENT: TOOL POOL AND SELECTION

To inject domain priors into the model, we give $F$ access to external tools, using directly $\Psi_M$ as $F$. The latter contribute to creating visual proxies for the given query, visualizing potential outcomes of the user intended modification. To achieve this, we instantiate two type of tools: web-search of exemplars (i) knowledge acquisition (via web search exemplars) In particular, external tools contribute (i) knowledge acquisition (via web search exemplars) to clarify ambiguous intent and (ii) pixel-level hypotheses (via image editing model) to instantiate requested modifications. These tools create visual exemplars which disambiguate queries and improve subsequent retrieval. In the following, we first describe the tools and how they are selected and applied in our framework.

**Set of tools.** We consider the set of tools {search, edit, none} with details described in Appendix A.1. Concretely, search issues a context-preserving query to a Web Search API to obtain a high-quality exemplar, while edit uses an image-editing model to generate a hypothesized target-like variant of $I_r$ guided by $T_m$. Both return a *tool-generated image* $I_{\text{tool}}$ that acts as a visual proxy. The option none, instead, considers the target caption as descriptive enough to be used as input for the retrieval module.

Figure 1: An overview of our model. An MLLM processes the reference image and the manipulation text with a tool-augmented reflective CoT to generate a target image description (and, when needed, a tool decision). The selected tool produces a visual proxy that refines the description, and a vision–language model performs image retrieval to obtain the final output.

**Selection via reflective CoT.** Given $(I_r, T_m)$, a tool-augmented chain-of-thought prompt jointly proposes an initial target image description $T_t^{(0)}$, a tool decision $a \in A$, and the tool instruction $\theta_a$ as follows:

$$\left(T_t^{(0)}, a, \theta_a\right) = \Psi_M(p_{\text{tool}} \circ I_r \circ T_m), \qquad a \in \{\texttt{edit}, \texttt{search}, \texttt{none}\}, \qquad (2)$$

where $p_{\text{tool}}$ is the selection prompt. For $\texttt{search}$, $\theta_a$ is a normalized, context-preserving query that retains the stable object/attributes from the reference image while adding only the requested modification from $T_m$ (*e.g.,* "black crew-neck T-shirt with a large lightsaber print" in Figure 1). For $\texttt{edit}$, $\theta_a$ is a concise edit query that explicitly states actions and explicit preservation constraints, while $T_m$ is a concise, executable instruction that differs from $T_m$ because the editor requires explicit operations and explicit preservation constraints rather than a brief, reference-dependent request (*e.g.,* "Change the front graphic to lightsaber, preserve black color and crew neckline"). If $a = \texttt{none}$, no external guidance is required and $T_t^{(0)}$ is used for target retrieval (Eq. 9).

**Tool use.** When $a \neq \texttt{none}$, the chosen tool $\Phi_{\text{tool}}^{(a)}$ produces a proxy image:

$$I_{\text{tool}} = \Phi_{\text{tool}}^{(a)}(I_r, T_m; \theta_a). \qquad (3)$$

Details on the execution can be found in the appendix, with both tools following Algorithm 1 with specialized routines (Algorithms 2, 3). We adopt a cache-based design to reduce the computational cost (please refer to Appendix A.3 for more details). The output of this stage is $T_t^{(0)}$ and the optional $I_{\text{tool}}$ with the associated metadata in case of web search.

### 3.3 REFINEMENT, ITERATION, AND REVISED SCORING

Given $\{I_r, T_m\}$ and $I_{\text{tool}}$, our goal is to produce a target description that is faithful to the intended edit while leveraging the proxy as a visual prior for retrieval. In the following, we detail how the description is refined, how this can be iterated, and how the final score uses the visual proxy.

**Target description refinement.** In case $a \neq \texttt{none}$, a refinement prompt $p_{\text{ref}}$ is used to instruct $\Psi_M$ for reflective chain-of-thought over the original reference image $I_r$, applying the modifications in $T_m$ ("Manipulation Text") while selectively incorporating evidence from $I_{\text{tool}}$ ("Tool Visual Proxy"). The refined target description $T_t$ is obtained as

$$T_t = T_t^{(1)} = \Psi_M(p_{\text{ref}} \circ I_r \circ T_m \circ I_{\text{tool}}), \qquad (4)$$

falling back to $T_t^{(1)} = T_t^{(0)}$ when $a = \texttt{none}$. The prompt enforces an extraction policy that enumerates preserved content (*e.g.,* category, shape, color, material), lists edits with attribute-level explicit values, and ignores proxy-specific distractors (*e.g.,* logos, extra patterns, lighting, or background) based on editing intention from $T_m$ For additional details, please refer to Appendix A.4.

Note that, while we set $T_t = T_t^{(1)}$, the model can use the collected evidence to re-iterate the selection and refinement process for multiple steps $k = 0, \dots, K$, where in each step it updates the visual

proxy and target image description. We set $K=2$ by default to preserve efficiency (details in Section 4.3), but we show results for multiple iterations in Figure 4.

**Composed Image Retrieval.** Given the final target image description $T_t$ and the optional tool-generated image $I_{\text{tool}}$, we perform retrieval from the database $\mathcal{D}$ using frozen CLIP encoders $\Psi_I$ and $\Psi_T$. The retrieved target image $I_t$ is obtained by maximizing a composite similarity score:

$$I_t = \arg\max_{I \in \mathcal{D}} \left( \frac{\Psi_I(I)^\top \Psi_T(T_t)}{\|\Psi_I(I)\| \, \|\Psi_T(T_t)\|} + \mathbb{I}_{\text{tool}} \frac{\Psi_I(I)^\top \Psi_I(I_{\text{tool}})}{\|\Psi_I(I)\| \, \|\Psi_I(I_{\text{tool}})\|} \right) \tag{5}$$

where $\mathbb{I}_{\text{tool}}$ is an indicator function that is 1 if a tool-generated visual proxy $I_{\text{tool}}$ is used and 0 otherwise. When no tool is invoked ($\mathbb{I}_{\text{tool}} = 0$), this equation simplifies to a direct text-to-image retrieval based on $T_t$, as in Eq. equation 1.

Note that the whole pipeline is fully modular and both $\Psi_M$ and the retriever can be replaced without affecting each other. Moreover, additional tools could be included to widen the expressivity of the model. Finally, the design of `TaCIR` is human-interpretable as all reasoning steps and target captions are expressed via language and, optionally visual examples.

## 4 EXPERIMENTS

**Datasets and metrics.** We evaluate on four standard CIR benchmarks: CIRR Liu et al. (2021) (*i.e.,* natural images; known false negatives), CIRCO Baldrati et al. (2023) (*i.e.,* multiple ground truths per query to mitigate false negatives), FashionIQ Wu et al. (2021) (*i.e.,* fine-grained fashion attribute edits), and GeneCIS Vaze et al. (2023) (*i.e.,* compositional retrieval over object/attribute variants built from MS-COCO Lin et al. (2014b) and VAW Pham et al. (2021)). We follow each benchmark's official protocol: report Recall@k (R@k) for CIRR, GeneCIS, and FashionIQ; mean average precision (mAP@k) for CIRCO due to multiple ground truths; and additionally Recall$_{\text{Subset}}$@k for CIRR to assess reasoning within the constrained candidate set. Further dataset statistics and evaluation details are provided in the Appendix A.8.

**Baselines.** We compare `TaCIR` against a range of widely benchmarked ZS-CIR methods, grouped into textual inversion (training-dependent) and training-free approaches. Among textual inversion baselines, we include: (1) **Pic2Word** Saito et al. (2023), which maps reference image features to pseudo-word tokens; (2) **SEARLE** Baldrati et al. (2023), which augments pseudo-word tokens with GPT-generated captions Brown et al. (2020); (3) **Context-I2W** Tang et al. (2024), which selectively maps text-relevant visual information from the reference image; (4) **LinCIR** Gu et al. (2024), which employs subject-masking in caption space to boost training efficiency; (5) **IP-CIR** Li et al. (2025)[1] and **CIG** Wang et al. (2025) use diffusion models to synthesize visual proxies for CIR: we report their published numbers when applied on LinCIR; and (6) **PrediCIR** Tang et al. (2025a), which predict the target image feature by a world model during inference.

For training-free baselines, we evaluate: (1) **CIReVL** Karthik et al. (2024), a two-stage framework where a pre-trained image captioner first generates a reference image caption, followed by an LLM that composes a target description; (2) **OSrCIR** Tang et al. (2025b), the first one-stage reflective CoT reasoning method for ZS-CIR, and (3) **OSrCIR***, which adapts OSrCIR by using the same MLLM as `TaCIR`, isolating the impact of model architecture.

To ensure fair comparison, we exclude ensemble methods such as LDRE Yang et al. (2024) as these introduce substantial computational overhead during inference for each query. All methods are benchmarked across three backbone architectures (ViT-B/32, ViT-L/14, ViT-G/14)Radford et al. (2021); Ilharco et al. but focus primarily on ViT-L/14 for baseline comparisons, as widely adopted in the literature Saito et al. (2023); Tang et al. (2024; 2025b;a).

**Implementation Details.** The default MLLM used in `TaCIR` is GPT-4.1 Achiam et al. (2023), while we also perform ablations with GPT-4o, O3, Gemini-2.5 and open-source MLLMs including LLaVA Liu et al. (2024) and Qwen2.5-VL Wang et al. (2024). GPT APIs are used with a temperature setting of 0, while all other parameters remain at their default values. The retrieval module performs all computations on a single NVIDIA H100 GPU. For the CLIP-based ViT variants Dosovitskiy (2020), we adopt weights from the official CLIP implementation Radford et al. (2021) while using

---

[1]IP-CIR does not report GeneCIS or ViT-L results on CIRCO/CIRR; where unavailable, we include CIG.

Table 1: **Comparison on CIRCO and CIRR Test Data.** On CIRCO, `TaCIR` significantly outperforms even adaptive methods across retrieval models, while it achieves competitive results on CIRR despite the noise in the benchmark. Grey lines represent the training-free ZS-CIR methods. OSrCIR* uses the GPT4.1. **Bold** and '_' denote the best and second-best result, respectively.

| CIRCO + CIRR → | | CIRCO | | | | CIRR | | | | | |
|---|---|---|---|---|---|---|---|---|---|---|---|
| Metric | | mAP@k | | | | Recall@k | | | Recall$_{Subset}$@k | | |
| Arch | Method | k=5 | k=10 | k=25 | k=50 | k=1 | k=5 | k=10 | k=1 | k=2 | k=3 |
| ViT-B/32 | SEARLE | 9.35 | 9.94 | 11.13 | 11.84 | 24.00 | 53.42 | 66.82 | 54.89 | 76.60 | 88.19 |
| | CIReVL | 14.94 | 15.42 | 17.00 | 17.82 | 23.94 | 52.51 | 66.00 | 60.17 | 80.05 | 90.19 |
| | OSrCIR | 18.04 | 19.17 | 20.94 | 21.85 | 25.42 | 54.54 | 68.19 | 62.31 | 80.86 | 91.13 |
| | OSrCIR* | 18.49 | 19.71 | 21.56 | 22.33 | 25.91 | 55.02 | 68.73 | 62.78 | 81.25 | 91.48 |
| | TaCIR | 21.02 | 22.35 | 24.71 | 25.60 | 28.93 | 58.19 | 71.12 | 65.27 | 83.74 | 93.58 |
| ViT-L/14 | Pic2Word | 8.72 | 9.51 | 10.64 | 11.29 | 23.90 | 51.70 | 65.30 | - | - | - |
| | SEARLE | 11.68 | 12.73 | 14.33 | 15.12 | 24.24 | 52.48 | 66.29 | 53.76 | 75.01 | 88.19 |
| | LinCIR | 12.59 | 13.58 | 15.00 | 15.85 | 25.04 | 53.25 | 66.68 | 57.11 | 77.37 | 88.89 |
| | +CIG | 12.84 | 13.77 | 15.25 | 16.12 | 26.17 | 54.94 | 67.64 | 58.00 | 77.86 | 89.34 |
| | Context-I2W | 13.04 | 14.62 | 16.14 | 17.16 | 25.60 | 55.10 | 68.50 | - | - | - |
| | PrediCIR | 15.70 | 17.10 | 18.60 | 19.30 | 27.20 | 57.00 | 70.20 | - | - | - |
| | CIReVL | 18.57 | 19.01 | 20.89 | 21.80 | 24.55 | 52.31 | 64.92 | 59.54 | 79.88 | 89.69 |
| | OSrCIR | 23.87 | 25.33 | 27.84 | 28.97 | 29.45 | 57.68 | 69.86 | 62.12 | 81.92 | 91.10 |
| | OSrCIR* | 24.36 | 25.98 | 28.62 | 29.81 | 29.93 | 58.22 | 70.41 | 62.66 | 82.43 | 91.47 |
| | TaCIR | 27.38 | 28.96 | 31.62 | 32.71 | 33.04 | 61.38 | 73.72 | 65.61 | 85.50 | 93.85 |
| ViT-G/14 | LinCIR | 19.71 | 21.01 | 23.13 | 24.18 | 35.25 | 64.72 | 76.05 | 63.35 | 82.22 | 91.98 |
| | +CIG | 20.64 | 21.90 | 24.04 | 25.20 | 36.05 | 66.31 | 76.96 | 64.94 | 83.18 | 91.93 |
| | +IP-CIR | 25.70 | 26.64 | 29.09 | 30.13 | 35.37 | 64.70 | 76.15 | 62.58 | 81.74 | 91.35 |
| | PrediCIR | 23.70 | 24.60 | 25.40 | 26.00 | 37.00 | 66.10 | 77.90 | - | - | - |
| | CIReVL | 26.77 | 27.59 | 29.96 | 31.03 | 34.65 | 64.29 | 75.06 | 67.95 | 84.87 | 93.21 |
| | OSrCIR | 30.47 | 31.14 | 35.03 | 36.59 | 37.26 | 67.25 | 77.33 | 69.22 | 85.28 | 93.55 |
| | OSrCIR* | 31.05 | 31.82 | 35.88 | 37.41 | 37.82 | 67.91 | 78.02 | 69.79 | 85.71 | 93.78 |
| | TaCIR | 34.28 | 35.22 | 39.41 | 40.68 | 40.72 | 71.06 | 80.95 | 72.06 | 87.89 | 95.04 |

OpenCLIP Ilharco et al. for ViT-G/14. Performance metrics are averaged across three trials to ensure reliability. For tools, we use Google's Programmable Search API as the default search backend and OpenAI's `gpt-image-1` as the default image editor.

## 4.1 QUANTITATIVE AND QUALITATIVE RESULTS

Our main quantitative experimental results are presented in Tables 1, 2, and 3, while Figures 2 and 3 show qualitative comparisons between our model and the baseline OSrCIR.

In Table 1, we show the comparison results for the CIRCO and CIRR datasets, which evaluate our model's capability in foreground and background differentiation as well as fine-grained image editing through object and scene manipulation tasks. Performances are evaluated on the hidden test sets of CIRCO and CIRR, accessible via the submission servers Baldrati et al. (2023); Saito et al. (2023). For all different CLIP-based ViT variants for retrieval, our approach significantly outperforms existing methods, including both training-free and textual inversion. For instance, on the default ViT-L/14 in CIRCO, which contains clean annotations of manipulation text with multiple target images, our method achieves a mAP5 of 27.38%, notably surpassing the 24.36% obtained by the best training-free baseline (OSrCIR*) and far above the 12.49% achieved by the SoTA textual inversion method (PrediCIR). The average performance rises to 30.17% versus 27.19% for OSrCIR*. Furthermore, in CIRR, where the manipulation text is less explicit and noisier, our method shows a 3.19% average improvement over OSrCIR* on ViT-L/14, with similar gains across other backbones and consistent improvements on the subset metric, indicating that optional tool consultation with a visual proxy enables the model to resolve implicit intent that is difficult for text-only target descriptions.

Qualitatively, as illustrated in Figure 2, `TaCIR` generates visual proxies that explicitly include intention-relevant attributes and context, so the retriever matches targets sharing those cues (*e.g.,* rendering a brown dog with a chain-link fence) guides the match to that background. Compared with OSrCIR, our `TaCIR` retain the "fence" (Row 1), the "vegetation" (Row 2), and the "puppy cupped in hands" interaction (Row 3), preserving fine-grained details crucial for alignment.

We further evaluate our model's capability on object and attribute composition using the GeneCIS dataset, with the results detailed in Table 2. Unlike CIRCO and CIRR, GeneCIS uses single-word manipulation texts with varied interpretations depending on the task, such as focusing on or changing a specific attribute or object. Consequently, user intent is often abstract and ambiguous, requiring

Table 2: Results on **GeneCIS** averaged over "Focus Attribute", "Change Attribute", "Focus Object", and "Change Object". Full table in Appendix A.5.

| Backbones | Methods | R1 | R2 | R3 |
|---|---|---|---|---|
| ViT-B/32 | SEARLE | 14.4 | 25.3 | 35.4 |
| | CIReVL | 15.8 | 26.8 | 36.8 |
| | OSrCIR | 17.4 | 29.1 | 39.0 |
| | OSrCIR* | 17.9 | 29.6 | 39.6 |
| | **TaCIR** | **19.9** | **32.0** | **42.2** |
| ViT-L/14 | SEARLE | 14.4 | 25.3 | 34.9 |
| | LinCIR | 12.2 | 22.8 | 32.4 |
| | +CIG | 13.6 | 24.4 | 33.6 |
| | PrediCIR | 16.6 | 26.7 | 35.8 |
| | CIReVL | 15.8 | 27.1 | 36.3 |
| | OSrCIR | 17.9 | 29.0 | 38.7 |
| | OSrCIR* | 18.3 | 29.5 | 39.3 |
| | **TaCIR** | **20.5** | **32.0** | **42.0** |
| ViT-G/14 | LinCIR | 13.7 | 24.7 | 33.6 |
| | PrediCIR | 17.7 | 28.9 | 38.6 |
| | CIReVL | 17.4 | 29.8 | 39.5 |
| | OSrCIR | 19.6 | 32.2 | 42.5 |
| | OSrCIR* | 20.1 | 32.8 | 43.2 |
| | **TaCIR** | **22.1** | **35.4** | **45.9** |

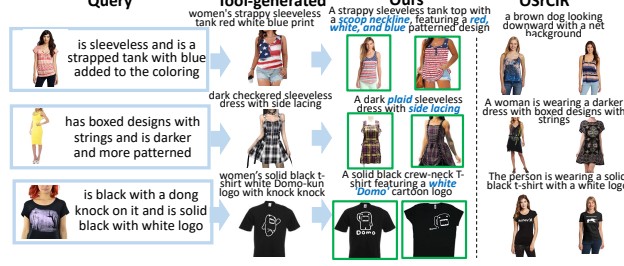

Figure 2: Object manipulation on CIRR.

Figure 3: Attribute manipulation on FashionIQ.

models to interpret intent precisely based on the reference image. For a fair comparison, we adopt the same output format as recent training-free work: for the "Focus" tasks, the MLLM is directed to retain the specified attribute or object, while for the "Change" tasks, it replaces the corresponding element. For the ViT-L/14 retrieval backbone, our method achieves an average R1 of 20.5%, improving over the best training-free baseline (OSrCIR*) by 2.20% and exceeding the leading textual inversion method by 5.13%. Similar improvements are observed for the other backbones: with ViT-B/32, the average R1 rises to 19.9% compared to 17.9% for OSrCIR*, and with ViT-G/14 it reaches 22.1% versus 20.1% for OSrCIR*. These results underscore the effectiveness of our tool-aware refinement in accurately resolving underspecified instructions on GeneCIS.

Lastly, Table 3 presents our model's performance on attribute manipulation tasks using the FashionIQ validation set, which requires accurate localization of specific fashion attributes (*e.g.*, style, color, pattern). The results show that `TaCIR` surpasses existing ZS–CIR models with the ViT–B/32 and ViT–L/14 backbones. For instance, on ViT–L/14, our method improves the average performance to 48.53%, exceeding the best training-free baseline OSrCIR* by 4.19% and the leading textual inversion method PrediCIR by 7.34%.

On ViT–G/14, our method achieves a notable 5.40% improvement over the best training-free baseline OSrCIR*, yet still trails the strongest textual inversion approach PrediCIR, whose training procedure is closely aligned with the CLIP retriever. This discrepancy likely reflects the advantage of retrieval–aligned supervision in the fashion domain, where CLIP's domain–specific semantics can be limited and fine-grained attribute manipulation intentions are harder to understand without such alignment. By contrast, in settings like CIRCO, where descriptions are more readily mapped into CLIP space, our training-free design brings larger gains. Thus, a promising future direction is to further enhance the alignment between the reasoning module and the retriever.

Qualitative comparison results of our method and the baseline method OSrCIR are presented in Figure 3. `TaCIR`, accurately localizes and edits attribute-relevant details of "a red–white–blue" strappy tank (Row 1), a "dark plaid sleeveless dress" with "side lacing" (Row 2), and a solid black "crew-neck tee" with a white "Domo" logo (Row 3).

## 4.2 ABLATION STUDY AND PERFORMANCE ANALYSIS

Similar to Karthik et al. (2024); Yang et al. (2024); Gu et al. (2024), we examine the contributions of core components in `TaCIR` on CIRCO and Fashion-IQ (Table 4). (1) **Models '2-6' assess the significance of key modules in `TaCIR`.** Removing all tool invocations (model '2') yields a 3.16% average drop compared to the full model (model '1'), underscoring the centrality of external augmentation. Disabling only web search (model '3') gives a 1.67% drop, highlighting the value of

Table 3: **Comparison on FashionIQ Validation Data.** TaCIR is able to significantly outperform adaptive methods across all sub-benchmarks, with its inherent modularity allowing for further simple scaling to achieve additional large gains. Grey lines represent the training-free ZS-CIR methods. OSrCIR* uses the GPT4.1. **Bold** and '_' denotes the best and second-best result, respectively.

| Fashion-IQ → | | Shirt | | Dress | | Toptee | | **Average** | |
|---|---|---|---|---|---|---|---|---|---|
| Backbone | Method | R@10 | R@50 | R@10 | R@50 | R@10 | R@50 | R@10 | R@50 |
| ViT-B/32 | SEARLE | 24.44 | 41.61 | 18.54 | 39.51 | 25.70 | 46.46 | 22.89 | 42.53 |
| | CIReVL | 28.36 | 47.84 | 25.29 | 46.36 | 31.21 | 53.85 | 28.29 | 49.35 |
| | OSrCIR | 31.16 | 51.13 | 29.35 | 50.37 | 36.51 | 58.71 | 32.34 | 53.40 |
| | OSrCIR* | 31.62 | 51.68 | 29.81 | 50.92 | 37.02 | 59.19 | 32.82 | 53.93 |
| | **TaCIR** | **34.92** | **55.22** | **33.14** | **54.12** | **40.37** | **62.42** | **36.14** | **57.25** |
| ViT-L/14 | Pic2Word | 26.20 | 43.60 | 20.00 | 40.20 | 27.90 | 47.40 | 24.70 | 43.70 |
| | SEARLE | 26.89 | 45.58 | 20.48 | 43.13 | 29.32 | 49.97 | 25.56 | 46.23 |
| | LinCIR | 29.10 | 46.81 | 20.92 | 42.44 | 28.81 | 50.18 | 26.28 | 46.49 |
| | + CIG | 28.66 | 47.20 | 21.27 | 43.98 | 29.83 | 50.28 | 26.59 | 47.15 |
| | Context-I2W | 29.70 | 48.60 | 23.10 | 45.30 | 30.60 | 52.90 | 27.80 | 48.90 |
| | PrediCIR | 31.80 | 52.00 | 25.40 | 49.50 | 33.10 | 55.40 | 30.10 | 52.30 |
| | CIReVL | 29.49 | 47.40 | 24.79 | 44.76 | 31.36 | 53.65 | 28.55 | 48.57 |
| | OSrCIR | 33.17 | 52.03 | 29.70 | 51.81 | 36.92 | 59.27 | 33.26 | 54.37 |
| | OSrCIR* | 33.71 | 52.61 | 30.12 | 52.36 | 37.41 | 59.85 | 33.75 | 54.94 |
| | **TaCIR** | **37.25** | **56.98** | **34.06** | **56.02** | **42.50** | **64.39** | **37.94** | **59.13** |
| ViT-G/14 | LinCIR | 46.76 | 65.11 | 38.08 | 60.88 | 50.48 | 71.09 | 45.11 | 65.69 |
| | + CIG | 47.35 | 66.68 | 39.71 | 60.93 | 50.69 | 71.39 | 45.92 | 66.34 |
| | + IP-CIR | 48.04 | 66.68 | 39.02 | 61.03 | 50.18 | 71.14 | 45.74 | 66.28 |
| | PrediCIR | **48.20** | **67.40** | **39.70** | **62.40** | **53.70** | **73.60** | **47.20** | **67.80** |
| | CIReVL | 33.71 | 51.42 | 27.07 | 49.53 | 35.80 | 56.14 | 32.19 | 52.36 |
| | OSrCIR | 38.65 | 54.71 | 33.02 | 54.78 | 41.04 | 61.83 | 37.57 | 57.11 |
| | OSrCIR* | 39.21 | 55.39 | 33.58 | 55.36 | 41.59 | 62.49 | 38.13 | 57.75 |
| | **TaCIR** | 44.18 | 60.41 | 39.62 | 61.76 | 46.77 | 67.28 | 43.52 | 63.15 |

extra-model knowledge for implicit intent. Removing image editing (model '4') forces text-only retrieval and gives a 1.46% drop, confirming the utility of the visual proxy. Skipping refinement (Model 5) or replacing the image-to-image term with the text prompt at retrieval (model '6') results in 1.17% and 1.36% drops, respectively, supporting both structured intent resolution and direct image-to-image matching. **(2) In models '7-8', we evaluate alternative solutions for key modules.** Replacing the default image generator (*i.e.,* gpt-image-1) with SDXL-Turbo Sauer et al. (2024) (model '7'), a highly efficient diffusion model requiring only ∼100ms per generation, causes only a minor 0.82% performance dip. This shows that TaCIR can be configured for high-speed inference with a minimal accuracy trade-off. An alternative retrieval strategy using combination of visual and textual features (model '8') instead of their independent scores results in a 1.39% performance drop, validating our design of using the synthesized image as an holistic query. **(3) In models '9-13', we analyze the impact of the choice of MLLM.** Open-source models, such as Qwen2.5-VL (model '10') and LLaVA (model '9'), achieve competitive results, but there remains an average performance gap of 1.46% and 0.97% compared to our full model with GPT-4.1 (model '1'). Notably, other API-based models like GPT-4o (model '12') perform comparably well, with only a 0.83% decline. The slightly larger drop for o3 (model '13') is attributed to its lower frequency of tool invocation for reasoning. Please refer to the Appendix A.7 for more ablation studies.

### 4.3 ANALYSIS

**Analysis of iterative tool-Use.** While our model invokes tools only once by default (*i.e.,* $K = 1$ in Sec. 3.3), we can allow multiple iterations over our tool selection and target description refinement pipeline. In Figure 4, we study the effect of increasing the number of iterations $K$ As the figure shows, performance increases *monotonically* w.r.t. the iterations, with diminishing returns. The largest jump is from $K = 1 \rightarrow 2$, while latency rises steadily. We therefore adopt $K = 2$ (one select&invoke followed by one refine) as the default, balancing performance and efficiency.

**Analysis of the Impact of Tool Calls.** Figure 5 isolates the impact of tool calls: relative to not using any tool (*No Tool*), allowing for a maximum of 1 select and invoke call (*Max 1 Call*) with early exit yields an average absolute gain of +3.3%. Always using tools, for any sample (*Always Use Tool*)

Table 4: Ablation study on CIRCO and FashionIQ.

| | CIRCO | | | Fashion-IQ | |
|---|---|---|---|---|---|
| Methods | k=5 | k=10 | k=25 | k=10 | k=50 |
| 1. Full model (GPT-4.1) | 27.38 | 28.96 | 31.62 | 37.94 | 59.13 |
| **Significance of key modules of TaCIR** | | | | | |
| 2. w/o tool invocation | 25.06 | 26.13 | 29.03 | 33.86 | 55.17 |
| 3. w/o searching | 26.47 | 27.92 | 30.39 | 35.39 | 56.49 |
| 4. w/o editing | 25.74 | 27.11 | 29.68 | 37.12 | 58.10 |
| 5. w/o target refinement | 26.24 | 27.73 | 30.18 | 37.01 | 58.03 |
| 6. w/o tool-generated image | 26.83 | 28.29 | 30.86 | 35.43 | 56.79 |
| **Alternative solutions for key modules** | | | | | |
| 7. SDXL Turbo | 26.97 | 28.38 | 30.97 | 36.61 | 58.02 |
| 8. feature combination | 26.23 | 27.68 | 30.11 | 36.13 | 57.95 |
| **Impact of different MLLMs** | | | | | |
| 9. LLaVA | 26.09 | 27.37 | 30.37 | 36.35 | 57.54 |
| 10. Qwen2.5-VL | 26.41 | 27.99 | 30.65 | 36.97 | 58.16 |
| 11. Gemini-2.5 | 26.25 | 27.83 | 30.49 | 36.81 | 58.08 |
| 12. gpt-4o | 26.55 | 28.13 | 30.79 | 37.11 | 58.30 |
| 13. o3 | 26.08 | 27.66 | 30.32 | 36.64 | 57.83 |

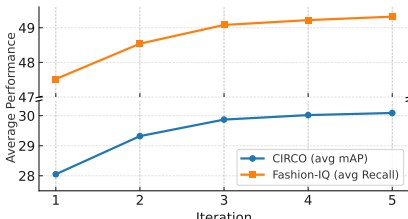

Figure 4: Improvement with iterations.

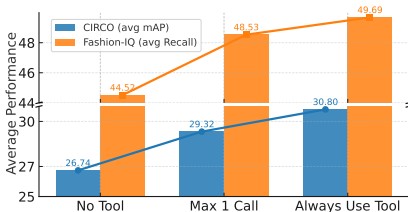

Figure 5: Tool calls effect.

achieves 4.16% average improvement confirming that using tools improve accuracy. Nevertheless, a selective single-call policy captures most of the benefit at modest cost.

**Analysis of Failure Cases.** To gain insights into failure cases of TaCIR, we analyzed 300 FashionIQ validation failures (ViT-G/14). As shown in Figure 6, we identify two main issues: (1) *Missing discriminative reference attributes* (67%). Prompts or tool text omit key cues from the reference (*e.g.,* color, silhouette, style), so the retriever prefers common but wrong variants. For example, dropping the *gray* tone and pocket type

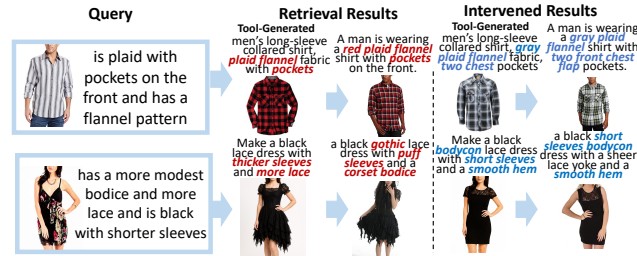

Figure 6: Visualization of common failure cases in the FashionIQ (top-2 retrieved results).

returns a red plaid shirt; adding "*gray* plaid flannel, *two front chest flap pockets*" fixes it (Row 1). (2) *Silhouette under-specification* (26%). Edits like "more lace" ignore shape constraints, resulting discouraged results. Making shape explicit(*e.g.,* "*bodycon* with *short sleeves* and a *smooth hem*") keeps the original silhouette and improves retrieval (Row 2).

**Efficiency Analysis.** Our approach improves over the best training-free baseline OSrCIR* by 2.20% to 4.16% across four CIR tasks while remaining interactive at $\sim 0.95$s per query, which contributes to our lightweight caching strategy. This latency is $\sim 1.6\times$ OSrCIR ($\sim 0.6$s) yet slightly below CIReVL ($\sim 1.0$s), and is achieved without task-specific training. Notably, even a one-iteration setting outperforms OSrCIR* by $\sim 2.41\%$ on average at comparable cost (*i.e.,* $\sim 0.75$s). Compared to textual-inversion methods, our performance surpasses them without training, but inference remains $\sim 48\times$ slower. As API calls account for 95% of inference time, faster APIs or improved tool selection could further reduce latency. For further details, please refer to our Appendix A.6.

## 5 CONCLUSION

In this paper, we present TaCIR, a tool-augmented, reflective reasoning agent for training-free ZS-CIR that jointly processes visual and textual inputs and consults external tools to resolve implicit manipulation intent. By acquiring external knowledge when needed and instantiating edits as a synthesized visual proxy, our approach reduces information loss common to text-only two-stage pipelines and aligns better with image–image retrieval. Across four diverse CIR benchmarks, TaCIR generalizes well and consistently outperforms prior training-free and textual-inversion methods, while maintaining competitive inference latency, with a two-iteration design further provides a favorable accuracy–efficiency trade-off. These findings show how an agentic system can provide advantages in compositional image retrieval. The pipeline is modular and future works may further improve the latter by using newer MLLMs, VLMs, and expanding the set of tools.

## REPRODUCIBILITY STATEMENT

We took several steps to ensure our results can be independently reproduced. The full method—selection, tool invocation, refinement, and fused retrieval, which is specified in Sec. 3, with algorithmic procedures and prompt templates provided verbatim in the appendix (tool pool, cache policy, and pseudocode for search/edit backends). Implementation details covering model/backbone choices (CLIP variants), tool backends (Google Search API, gpt-image-1), decoding parameters, seeds, and hardware are reported in Sec. 4 and expanded in the appendix (including ablation protocols and sensitivity analyses). Dataset usage follows official benchmarks, with splits, preprocessing, and evaluation metrics summarized in Sec. 4 and detailed in the appendix. An anonymized repository in the supplementary materials includes environment specifications, configuration files, inference/evaluation scripts, and a minimal working example with sample data; cached artifacts and provenance metadata are provided to stabilize runs involving external tools. Large language models were used both as a module of our method and to aid writing; roles and configurations are documented in Sec. 4 and the appendix.

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

## A  APPENDIX

### A.1  TOOL POOL

We expose a compact tool pool to the *tool-augmented CoT prompt* so that the MLLM can select external assistance only when beneficial. The pool comprises (i) a **Search API** that returns a set of context-preserving visual exemplars (online images with titles) for clarifying specialized terminology or comparative modifiers, and (ii) an **Image Editing** operator that synthesizes a *tool-generated image* as a concrete visual proxy when the intended transformation is too intricate to specify reliably in text. A **No-Tool** option (`none`) is retained for simple, unambiguous edits, preserving efficiency and avoiding unnecessary calls.

### A.2  DETAILS OF TOOL SELECTION PROMPT

Given a reference image $I_r$ and manipulation text $T_m$, zero-shot CIR (ZS-CIR) requires disambiguating user intent that may be ambiguous, comparative, or domain-specific and beyond the frozen knowledge of the MLLM. To address this, we design a *tool-augmented chain-of-thought (CoT) prompt* $p_{\text{tool}}$ that guides the MLLM to: (i) generate an initial target image description $T_t^{(0)}$, (ii) decide whether an external tool is needed, and (iii) if so, emit a minimal, executable instruction $\theta_a$ for tool invocation. The tool choice variable $a \in \{\text{search}, \text{edit}, \text{none}\}$ determines whether to call a Search API, invoke an image editing tool, or proceed without tool use, respectively.

While conventional MLLM-based, training-free ZS-CIR methods directly generate a target image description from the reference image and manipulation text, they are constrained by the frozen knowledge of the MLLM and often struggle with implicit intent or domain-specific details. To address this, we propose an adaptive framework that allows the MLLM to *optionally* consult external

tools to enhance reasoning. As shown in Figure 1, the MLLM first processes the reference image $I_r$ and manipulation text $T_m$ and decides whether to invoke an external tool to obtain a **tool-generated image** that serves as a visual proxy. The final target description $T_t$ is then reasoned from the original inputs together with this proxy.

Formally, given an MLLM $\Psi_M$, the *tool-augmented CoT prompt* jointly emits a preliminary target description and a tool decision:

$$\left(T_t^{(0)}, a, \theta_a\right) = \Psi_M(p_{\text{tool}} \circ I_r \circ T_m), \qquad a \in \{\text{edit}, \text{search}, \text{none}\}, \qquad (6)$$

where $p_{\text{tool}}$ is a chain-of-thought prompt and $\theta_a$ denotes the minimal *tool instruction* (*e.g.,* a search query or an edit instruction) to be executed if $a \neq \text{none}$. If $a = \text{none}$, no external guidance is required. We directly use the initial target image description for retrieval as:

$$T_t = T_t^{(0)} = \Psi_M(p_{\text{tool}} \circ I_r \circ T_m). \qquad (7)$$

Otherwise, we invoke the chosen tool $\Phi_{\text{tool}}^{(a)}$ to produce a tool-generated image $I_{tool}$ as follows:

$$I_{tool} = \Phi_{\text{tool}}^{(a)}(I_r, T_m; \theta_a). \qquad (8)$$

This visual proxy supplies extra-model knowledge and concrete visual evidence, serving as additional context for $\Psi_M$. We then refine the target description with a concise refinement prompt $p_{\text{ref}}$:

$$T_t = T_t^{(1)} = \Psi_M(p_{\text{ref}} \circ I_r \circ T_m \circ I_{tool}). \qquad (9)$$

In practice, prompts follow a task-agnostic structure: $I_r$ is introduced as "`Original Image Context`", $T_m$ as "`Manipulation Text`", and the tool-generated image $I_{\text{tool}}$ as "`Tool Visual Proxy`". This design foregrounds the advantage of tool use: $I_{\text{tool}}$ injects extra-model knowledge and provides a concrete visual anchor that disambiguates implicit intent and encodes domain-specific constraints, thereby improving the faithfulness of $T_t$ and its alignment with retrieval. When $\Psi_M$ predicts $a \neq \text{none}$, it *also supplies the executable tool instruction $\theta_a$*, which directs the generation of $I_{\text{tool}}$ prior to the final refinement step.

**Tool Selection.** The tool-selected reflective chain-of-thought (CoT) prompt $p_{\text{tool}}$ unifies intent understanding and tool selection in one stage. Given a reference image $I_r$ and manipulation text $T_m$, the MLLM first summarizes intent-relevant attributes from $I_r$, then reasons through the manipulation described in $T_m$, articulating how each modification is interpreted and prioritized.

Critically, the model then reflects on whether ambiguity, implicit intent, or domain-specific gaps remain unresolved. If further evidence is needed, it selects a tool ($a \in \{\text{search}, \text{edit}, \text{none}\}$), justifying its choice in context. For $a \neq \text{none}$, the model emits a minimal, executable instruction, either a context-preserving search query or an image edit script—explicitly formatted for the invocation module.

If no tool is required, the initial target description $T_t^{(0)}$ is used for retrieval; otherwise, the selected tool and instruction guide the generation of a visual proxy for downstream augmentation. This approach injects external knowledge or visual evidence only when necessary, improving robustness to ambiguity and domain specificity, while preserving efficiency and interpretability. Specifically:

**Prompt Structure and Reasoning Process.**

- **Original Image Description:** The MLLM first generates a detailed, intent-relevant summary of $I_r$, explicitly capturing all key objects, attributes, colors, styles, and scene elements while omitting irrelevant background content. This provides the contextual basis for all subsequent reasoning.

- **Thoughts:** The model then articulates its internal reasoning about the manipulation intent, detailing how each modification was interpreted and prioritized, which semantic cues in $T_m$ were most relevant, and how these influenced the generated description.

- **Reflections and Tool Decision:** The MLLM reflects on whether the manipulation can be adequately addressed using its current knowledge. If ambiguity, specialized terminology, or complex visual edits are present, it evaluates whether to invoke the `search` tool (for context-preserving exemplars) or the `edit` tool (for intricate visual transformations). The rationale for tool selection is made explicit, and if no tool is needed, $a = \text{none}$ is selected.

You are an image description expert. You are provided with an original image and a manipulation text. Note that manipulation text with multiple intents describes changes from the original image to a target image. Your goal is to generate a concise, precise, and clear target image description that reflects the manipulation intents while preserving as much of the original image content as possible.

## Available Tools:
- You can use the following tools when encountering ambiguities or limited knowledge:

### Search API
- Leverages Google API to provide actual images with titles for visual reference
- Call when you need visual examples or domain-specific clarification
    - Input: Query that includes original context + modification
    - Output: Actual Google images with titles that match both original style and modification

### Image Editing Tool
- Call when transformation is too visually complex for text description
    - Input: Reference image and detailed edit instruction
    - Output: Edited image preserving original style elements

## Guidelines on generating the Original Image Description
- Ensure the original image description is thorough, capturing all visible objects, attributes, and elements.
- The original image description should be as accurate as possible, reflecting the content of the image.

## Guidelines on generating the Thoughts
- In your Thoughts, explain your understanding of the manipulation intents and how you formulated the target image description.
- Provide insight into how you interpreted the manipulation intent in detail in the manipulation text.
- Discuss how the manipulation intent influenced which elements of the original image you focused.

## Guidelines on generating the Reflections
- State your interpretation of the manipulation
- Decide if tools needed (Yes/No and why)
- If comparative terms → do you need visual reference?
- If complex visuals → do you need editing tool?

#### Guidelines on generating Tool Usage
Clearly specify the tool(s) called, including the rationale behind calling each tool.
- Clearly state your queries in the following format for each tool:
- Searching API query: <search>your query here</search>
- Image editing tool query: <edit>your edit instruction here</edit>e
- If no tools are necessary, explicitly state "None".

## Guidelines on generating Target Image Description
- The target image description you generate should be complete and can cover various semantic aspects.
- The target image description only contains the target image content and needs to be as simple as possible. Minimize aesthetic descriptions as much as possible.

## On the input format:
…

Figure 7: The complete template of our tool-selection reflective Chain-of-Thought process for Training-free ZS-CIR.

- **Executable Tool Instruction** ($\theta_a$)**:** When a tool is chosen, the model emits a minimal instruction: for `search`, this is a context-preserving query string combining the original item's context with the requested modification (e.g., "formal evening dress darker than navy"); for `edit`, a concise script describing the required transformation relative to $I_r$ (e.g., "add bell sleeves; keep color and silhouette unchanged"). The instruction is designed for direct use by the tool-invocation module.

- **Target Image Description:** Finally, the MLLM produces an initial target description $T_t^{(0)}$ that reflects all manipulation intents from $T_m$ while retaining unedited content from $I_r$. The output is required to be concise, explicit, and directly interpretable by a downstream retrieval model, with careful handling of comparative expressions and precise attribute terms.

**Guidelines for Tool Usage.** The prompt encourages use of the Search API for clarification of ambiguous, comparative, or domain-specific terms, and use of Image Editing for transformations that are visually complex or hard to express textually. The `none` option is reserved for simple, unambiguous edits. The overall design enables the MLLM to self-assess its limitations, invoke external tools only when beneficial, and produce precise, retrieval-compatible target descriptions. When no tool is required, the system reverts to the efficient baseline path, but adaptively injects extra-model knowledge or visual grounding when needed, thereby ensuring both accuracy and flexibility in ZS-CIR.

### A.3 DETAILS OF TOOL INVOCATION PROCESS

Given the selection output $(T_t^{(0)}, a, \theta_a)$, the invocation module executes the chosen action and prepares the signal for refinement (Algorithm 1). When $a = $ `search`, the module issues a context-preserving query using a *cache-first* policy (Algorithm 2): queries are deterministically hashed; cache hits return a local visual exemplar and metadata with zero network I/O, while misses fetch the *top* result, verify integrity, optionally resize, and persist the artifact and index. When $a = $ `edit`, the module similarly applies a cache-first edit path keyed by a content–instruction hash (Algorithm 3); cache hits return the edited image immediately, otherwise the API output is decoded, verified, and stored with provenance. The resulting visual proxy (*e.g.,* search exemplar or edited image) is then fed back to the MLLM in a refinement prompt $p_{\mathrm{ref}}$ to produce the final $T_t$. By prioritizing cached artifacts and indexing all successful calls, the module substantially reduces latency and external calls while preserving the gains of tool-augmented reasoning.

**Tool Invocation.** After the tool-selection stage outputs $(T_t^{(0)}, a, \theta_a)$, the invocation module executes the chosen action and prepares the signal for the subsequent refinement round (Algorithm 1). The module first robustly decodes the MLLM payload by stripping wrapper tags (e.g., `<Response>...</Response>`, fenced code blocks) and parsing JSON fields for *Thoughts*, *Reflections*, *Tool Usage*, and the *Target Image Description*. It records the first-pass description and reasoning $(T_t^{(0)}, \mathcal{H}, \mathcal{R})$ for analysis. If no tool is requested ($a = $ `none`) or the iteration budget $K$ is reached, the module immediately returns $T_t^{(0)}$ for retrieval.

When $a = $ `search`, the module extracts a context-preserving query $\theta_a$ (sanitizing `<search>...</search>`) and invokes a cache-first visual search (Algorithm 2). Queries are deterministically hashed; a cache hit returns the local visual exemplar and metadata (title, source, domain) with *zero* network cost. On a miss, the system issues a single top-result request, downloads the image, verifies integrity, optionally resizes to a bounded resolution, and persists both the image and metadata into a structured index. The result is formatted as a compact "Visual Reference" (text header plus local image path) and fed back to the MLLM in the refinement prompt.

When $a = $ `edit`, the module sanitizes the edit script $\theta_a$ (removing `<edit>...</edit>`), verifies the availability of a local source image, and consults an edit cache keyed by a content–instruction hash (Algorithm 3). A cache hit returns the previously produced edited image. Otherwise, the module submits the request to the image-editing API, decodes the base64 response, verifies the image, and persists it along with provenance (original image, manipulation text, optional ground truth). The edited image serves as a tool-generated visual proxy and is passed to the MLLM during refinement.

For reliability and throughput, both search and edit paths use operation-level locks to prevent duplicate requests for the same query or edit instruction (Algorithms 2–3). The invocation interface returns the refined target description together with lightweight metadata (original/refined descriptions, thoughts, selected tool, and the emitted tool instruction) for inspection. Empirically, the cache substantially reduces latency and external calls—especially for repeated or semantically similar queries—thereby improving efficiency while preserving the quality gains of tool-augmented reasoning.

---

**Algorithm 1** Tool Invocation for Tool-Augmented ZS-CIR

---

**Input**: initial triplet from tool selection $(T_t^{(0)}, a, \theta_a)$, reference image path $I_r$, manipulation text $T_m$, optional target path $I_{\text{gt}}$, max iterations $K$, tools pool $\Phi_{\text{pool}}$
**Parameters**: search cache $\mathcal{C}_{\text{search}}$, edit cache $\mathcal{C}_{\text{edit}}$, operation locks $\mathcal{L}_{\text{search}}, \mathcal{L}_{\text{edit}}$
**Output**: final target description $T_t$, thoughts $\mathcal{H}$, reflections $\mathcal{R}$, tool usage record $\mathcal{U}$

1: $t \leftarrow 0$; $\texttt{used} \leftarrow \texttt{False}$; $I_{\text{curr}} \leftarrow I_r$
2: Initialize $\mathcal{H}, \mathcal{R}, \mathcal{U} \leftarrow \varnothing$; $T_t^{\text{orig}} \leftarrow \varnothing$
3: **while** $t \leq K$ **do**
4:    **if** $t = 0$ **then**   // parse first MLLM response (already produced by selection)
5:       $T_t^{\text{resp}} \leftarrow T_t^{(0)}$; $a^{\text{resp}} \leftarrow a$; $\theta^{\text{resp}} \leftarrow \theta_a$
6:    **else**   // refinement round after tool execution
7:       $(T_t^{\text{resp}}, a^{\text{resp}}, \theta^{\text{resp}}) \leftarrow \Psi_M(p_{\text{ref}} \circ I_r \circ T_m \circ I_{\text{curr}})$
8:    Extract and store *Thoughts* $\mathcal{H}_t$, *Reflections* $\mathcal{R}_t$, *Tool Usage* $\mathcal{U}_t$ from the JSON payload
9:    **if** $t = 0$ **then** $T_t^{\text{orig}} \leftarrow T_t^{\text{resp}}$; $\mathcal{H} \leftarrow \mathcal{H}_t$; $\mathcal{R} \leftarrow \mathcal{R}_t$
10:   **if** $\texttt{used} = \texttt{True}$ **and** $t > 0$ **then return** $T_t^{\text{resp}}, (\mathcal{H} \cup \mathcal{H}_t), (\mathcal{R} \cup \mathcal{R}_t), \mathcal{U}$
11:   **if** $(a^{\text{resp}} = \texttt{none})$ **or** $(t = K)$ **then return** $T_t^{\text{resp}}, \mathcal{H}, \mathcal{R}, \mathcal{U}$   // no tool or budget exhausted
12:   // execute exactly one tool based on $a^{\text{resp}}$ and minimal instruction $\theta^{\text{resp}}$
13:   **if** $a^{\text{resp}} = \texttt{search}$ **then**
14:      $I_{\text{ref}} \leftarrow \text{SEARCHINVOKE}(\theta^{\text{resp}}, \Phi_{\text{pool}}, \mathcal{C}_{\text{search}}, \mathcal{L}_{\text{search}})$
15:      **if** $I_{\text{ref}} \neq \varnothing$ **then**
16:         $I_{\text{curr}} \leftarrow I_{\text{ref}}$; $\texttt{used} \leftarrow \texttt{True}$; $\mathcal{U} \leftarrow \mathcal{U}_t$
17:      **end if**
18:   **else if** $a^{\text{resp}} = \texttt{edit}$ **then**
19:      $I_{\text{edit}} \leftarrow \text{EDITINVOKE}(I_{\text{curr}}, \theta^{\text{resp}}, \Phi_{\text{pool}}, \mathcal{C}_{\text{edit}}, \mathcal{L}_{\text{edit}}, I_{\text{gt}}, T_m)$
20:      **if** $I_{\text{edit}} \neq \varnothing$ **then**
21:         $I_{\text{curr}} \leftarrow I_{\text{edit}}$; $\texttt{used} \leftarrow \texttt{True}$; $\mathcal{U} \leftarrow \mathcal{U}_t$
22:      **end if**
23:   **end if**
24:   $t \leftarrow t + 1$
25: **end while**
26: **return** $T_t^{\text{orig}}, \mathcal{H}, \mathcal{R}, \mathcal{U}$

---

**Algorithm 2** SEARCHINVOKE: cache-first visual search with formatting

---

**Input**: context-preserving query $\theta$, tools pool $\Phi_{\text{pool}}$, search cache $\mathcal{C}_{\text{search}}$, lock $\mathcal{L}_{\text{search}}$
**Output**: local path to visual reference $I_{\text{ref}}$ (or $\varnothing$)

1: $h \leftarrow \text{HASH}(\theta)$
2: **if** $\mathcal{C}_{\text{search}}[h]$ exists **then return** $\mathcal{C}_{\text{search}}[h].\texttt{image\_path}$    *// cache hit: zero network cost*
3: Acquire $\mathcal{L}_{\text{search}}$; **defer** release
4: $R \leftarrow \Phi_{\text{pool}}.\text{SEARCH\_IMAGES\_WITH\_DOWNLOAD}(\theta, 1)$
5: **if** $R \neq \varnothing$ **and** $R[0].\texttt{image\_available} = \texttt{True}$ **then**
6:    $I_{\text{ref}} \leftarrow R[0].\texttt{local\_image\_path}$; $\text{UPDATECACHE}(\mathcal{C}_{\text{search}}, h, R[0])$
7:    **return** $I_{\text{ref}}$
8: **else**
9:    **return** $\varnothing$
10: **end if**

---

## A.4 DETAILS OF TARGET IMAGE DESCRIPTION REFINEMENT PROMPT

Given a reference image $I_r$, manipulation text $T_m$, and a tool-generated reference image $I_{\text{tool}}$ (from visual search or image editing), we design a refinement prompt $p_{\text{ref}}$ to synthesize the final target image description $T_t$. The prompt treats $I_{\text{tool}}$ strictly as auxiliary evidence to clarify the specific modification requested in $T_m$; it is not the target and must not introduce unrelated content or style drift. The objective is a concise, precise description that remains grounded in $I_r$ while applying only the necessary change indicated by $T_m$.

---

**Algorithm 3** EDITINVOKE: cached image editing/generation

---

**Input**: current image path $I_{\text{curr}}$, edit script $\theta$, tools pool $\Phi_{\text{pool}}$, edit cache $\mathcal{C}_{\text{edit}}$, lock $\mathcal{L}_{\text{edit}}$, optional $I_{\text{gt}}, T_m$
**Output**: local path to edited image $I_{\text{edit}}$ (or $\varnothing$)

1: $h \leftarrow \text{HASH}(I_{\text{curr}}, \theta)$
2: **if** $\mathcal{C}_{\text{edit}}[h]$ exists **then return** $\mathcal{C}_{\text{edit}}[h].\texttt{output\_path}$            *// cache hit*
3: **if** $\neg\text{EXISTS}(I_{\text{curr}})$ **then return** $\varnothing$
4: Acquire $\mathcal{L}_{\text{edit}}$; **defer** release
5: $P \leftarrow \Phi_{\text{pool}}.\text{EDIT\_IMAGE}(I_{\text{curr}}, \theta, \texttt{output\_dir} = \varnothing, \texttt{original\_manipulation} = T_m, \texttt{target\_image\_path} = I_{\text{gt}})$
6: **if** $P \neq \varnothing$ **then**
7:    $\text{UPDATECACHE}(\mathcal{C}_{\text{edit}}, h, P)$; $I_{\text{edit}} \leftarrow P$
8:    **return** $I_{\text{edit}}$
9: **else**
10:   **return** $\varnothing$
11: **end if**

---

**Prompt Structure and Reasoning Process.**

- **Original Image Description:** The MLLM first produces an intent-focused description of $I_r$, capturing salient objects, attributes, colors, styles, and scene elements while omitting irrelevant background content. This establishes the grounding context for refinement.

- **Tool-Generated Visual Evidence Description:** The model then describes $I_{\text{tool}}$ (search exemplar with title or edited variant) under a *strict extraction policy*: extract only the attribute relevant to $T_m$; explicitly ignore unrelated styles, patterns, logos, backgrounds, or accessories. For search exemplars, the title is quoted and non-essential differences are listed as ignored; for edited images, unintended artifacts are identified and ignored.

- **Thoughts:** The model (i) parses the manipulation intent from $T_m$; (ii) inventories elements from $I_r$ and marks them as [PRESERVE] or [MODIFY]; (iii) specifies exactly what is extracted from $I_{\text{tool}}$ and what is ignored; and (iv) explains the combination strategy that applies the minimal necessary change while preserving all other content.

- **Reflections:** A brief, three-part reflection is required: state the single extracted modification; enumerate ignored elements from $I_{\text{tool}}$; confirm preservation of unmodified content from $I_r$. This format enforces transparency and supports auditing.

- **Tool Invocation (if applicable).** If the selection stage outputs $(a, \theta_a)$ with $a \neq \texttt{none}$, the refinement prompt begins by *invoking* the chosen tool $\Phi_{\text{tool}}^{(a)}$ using $\theta_a$ (cf. Eq. 8). Invocation follows a cache-first policy. If invocation fails or the artifact is rejected, the prompt sets $a \leftarrow \texttt{none}$ and proceeds without a proxy. The prompt receives *only* the proxy and its minimal provenance tokens, never raw HTML or executable code.

- **Target Image Description:** The final description integrates the preserved content of $I_r$ with only the single modification clarified by $I_{\text{tool}}$. The output is concise and self-contained, using explicit comparative phrasing and precise domain terms suitable for a downstream retrieval model.

**Context Selection Principle.** The manipulation text $T_m$ is the authoritative signal for what changes; attributes not explicitly requested must be preserved. The tool-generated reference $I_{\text{tool}}$ serves only to *disambiguate* the requested change and must not introduce additional modifications.

The refinement prompt $p_{\text{ref}}$ delivers a controlled synthesis: $I_r$ provides grounding, $I_{\text{tool}}$ supplies narrowly scoped evidence for the requested modification, and $T_m$ governs what may change. This yields faithful, precise, and retriever-compatible target descriptions while avoiding over-transfer from tool evidence.

A.5   THE FULL TABLE OF GENECIS

In Table 5, we report the full table of GeneCIS results.

You are an image description expert. You are provided with an original image, a manipulation text, and a single tool-generated reference image (either a search result with title or an edited image) that demonstrates the intended changes. Note that manipulation text with multiple intents describes changes from the original image to a target image. Your goal is to generate a concise, precise, and clear target image description by intelligently selecting and combining context from both the original image and the tool-generated reference image, guided by the manipulation intent.

## Guidelines on generating the Original Image Description
- Ensure the original image description is thorough, capturing all visible objects, attributes, and elements.
- The original image description should be as accurate as possible, reflecting the content of the image.

## Guidelines on generating Tool-Generated Visual Evidence Description

- The Tool-Generated Visual Evidence description should accurately capture what you observe in the tool result image, similar in detail to the original image description:

## Guidelines on generating the Thoughts

In your Thoughts, follow this systematic analysis based on the manipulation text:

1. Parse Manipulation Intent (first sentence):
   - Quote the manipulation text and identify the specific change requested
   - Explicitly state: "This requests ONLY changing [X] while preserving EVERYTHING else"

2. Inventory Original Elements (second part):
   - List ALL elements from the original image
   - Mark each as [PRESERVE] or [MODIFY] based on manipulation text

3. Extract Tool Evidence WITH WARNINGS (third part):
   - State what ONE thing the tool provides for modification

4. Combination Strategy*(final part):
   - Explain: "I will take ONLY [one modification] from tool and preserve EVERYTHING else from original"

## Guidelines on generating the Reflections

 - Structure your reflection in exactly three components:

- Component 1: State the SINGLE modification extracted
- Component 2: List what you IGNORED from tool (most of it)
- Component 3: Confirm all preserved elements

   #### Guidelines on generating Tool Usage
   Clearly specify the tool(s) called, including the rationale behind calling each tool.
   - Searching API query: `<search>your query here</search>`
   - Image editing tool instruction: `<edit>your edit instruction here</edit>`
   - If no tools are necessary, explicitly state: `None`.

## Guidelines on generating Target Image Description
   - The target image description should be complete and cover various semantic aspects
   - Ensure the description is clear even without knowledge of the original image
   - Keep the description concise and simple, minimizing aesthetic details

## On the input format:
 …

Figure 8: The complete template of our tool-refinement reflective Chain-of-Thought process for Training-free ZS-CIR.

## A.6  MORE ANALYSIS ON EFFECTIVENESS AND EFFICIENCY

Our approach improves over the best training-free baseline OSrCIR* by 2.20% to 4.16% across four CIR tasks while remaining interactive at $\sim 0.95$s per query, which contributes to our lightweight

Table 5: **Comparison on GeneCIS Test Data.** `TaCIR` is able to significantly outperform adaptive methods across all GeneCIS sub-benchmarks, with its inherent modularity allowing for further simple scaling to achieve additional large gains. Grey lines represent the training-free ZS-CIR methods. OSrCIR* uses the GPT4.1. **Bold** and '_' denotes the best and second-best result, respectively.

| GeneCIS → | | Focus Attribute | | | Change Attribute | | | Focus Object | | | Change Object | | | Average |
|---|---|---|---|---|---|---|---|---|---|---|---|---|---|---|
| Backbone | Method | R@1 | R@2 | R@3 | R@1 | R@2 | R@3 | R@1 | R@2 | R@3 | R@1 | R@2 | R@3 | R@1 |
| ViT-B/32 | SEARLE | 18.9 | 30.6 | 41.2 | 13.0 | 23.8 | 33.7 | 12.2 | 23.0 | 33.3 | 13.6 | 23.8 | 33.3 | 14.4 |
| | CIReVL | 17.9 | 29.4 | 40.4 | 14.8 | 25.8 | 35.8 | 14.6 | 24.3 | 33.3 | 16.1 | 27.8 | 37.6 | 15.9 |
| | OSrCIR | 19.4 | 32.7 | 42.8 | 16.4 | 27.7 | 38.1 | 15.7 | 25.7 | 35.8 | 18.2 | 30.1 | 39.4 | 17.4 |
| | OSrCIR* | 19.8 | 33.2 | 43.3 | 16.9 | 28.1 | 38.7 | 16.1 | 26.3 | 36.2 | 18.7 | 30.7 | 40.1 | 17.9 |
| | **TaCIR** | **22.0** | **36.0** | **46.1** | **18.7** | **30.2** | **41.0** | **17.9** | **28.4** | **38.5** | **20.9** | **33.2** | **43.1** | **19.9** |
| ViT-L/14 | SEARLE | 17.1 | 29.6 | 40.7 | 16.3 | 25.2 | 34.2 | 12.0 | 22.2 | 30.9 | 12.0 | 24.1 | 33.9 | 14.4 |
| | LinCIR | 16.9 | 30.0 | 41.5 | 16.2 | 28.0 | 36.8 | 8.3 | 17.4 | 26.2 | 7.4 | 15.7 | 25.0 | 12.2 |
| | Context-I2W | 17.2 | 30.5 | 41.7 | 16.4 | 28.3 | 37.1 | 8.7 | 17.9 | 26.9 | 7.7 | 16.0 | 25.4 | 12.7 |
| | PrediCIR | 18.2 | 31.9 | 42.6 | 18.7 | 30.4 | 35.4 | 12.7 | 19.0 | 31.2 | 16.9 | 25.5 | 34.1 | 16.6 |
| | CIReVL | 19.5 | 31.8 | 42.0 | 14.4 | 26.0 | 35.2 | 12.3 | 21.8 | 30.5 | 17.2 | 28.9 | 37.6 | 15.9 |
| | OSrCIR | 20.9 | 33.1 | 44.5 | 17.2 | 28.5 | 37.9 | 15.0 | 23.6 | 34.2 | 18.4 | 30.6 | 38.3 | 17.9 |
| | OSrCIR* | 21.3 | 33.6 | 45.1 | 17.6 | 29.1 | 38.5 | 15.4 | 24.1 | 34.8 | 18.9 | 31.2 | 39.0 | 18.3 |
| | **TaCIR** | **23.6** | **35.9** | **47.4** | **19.6** | **31.6** | **41.3** | **17.6** | **26.6** | **37.6** | **21.0** | **33.7** | **41.8** | **20.5** |
| ViT-G/14 | LinCIR | 19.1 | 33.0 | 42.3 | 17.6 | 30.2 | 38.1 | 10.1 | 19.1 | 28.1 | 7.9 | 16.3 | 25.7 | 13.7 |
| | PrediCIR | 19.3 | 33.2 | 42.7 | 19.9 | 30.7 | 38.9 | 12.8 | 19.4 | 32.3 | 18.9 | 32.2 | 40.6 | 18.7 |
| | CIReVL | 20.5 | 34.0 | 44.5 | 16.1 | 28.6 | 39.4 | 14.7 | 25.2 | 33.0 | 18.1 | 31.2 | 41.0 | 17.4 |
| | OSrCIR | 22.7 | 36.4 | 47.0 | 17.9 | 30.8 | 42.0 | 16.9 | 28.4 | 36.7 | 21.0 | 33.4 | 44.2 | 19.6 |
| | OSrCIR* | 23.2 | 36.9 | 47.7 | 18.4 | 31.4 | 42.7 | 17.3 | 29.0 | 37.3 | 21.5 | 34.0 | 45.0 | 20.1 |
| | **TaCIR** | **25.4** | **39.7** | **50.6** | **20.4** | **34.0** | **45.0** | **19.3** | **31.2** | **40.1** | **23.4** | **36.8** | **47.8** | **22.1** |

Table 6: Comparison of Computational Cost.

| Model | LLM | Latency | GPU Memory | API Cost | Performance |
|---|---|---|---|---|---|
| Context-I2W | * | $\sim 0.02$s | 16 GB | $0 | 22.94 |
| CIReVL | GPT-3.5 | $\sim 1$s | 40 GB | $\sim$ $0.001 | 26.23 |
| OSrCIR | GPT-4o | $\sim 0.7 \pm 0.08$s | 16 GB | $\sim$ $0.004 | 32.27 |
| TaCIR | GPT-4o | $\sim 0.83 \pm 0.08$s | 16 GB | $\sim$ $0.011 | 36.18 |
| TaCIR(w/o cache) | GPT-4.1 | $\sim 1.38 \pm 0.08$s | 16 GB | $\sim$ $0.007 | 37.01 |
| TaCIR | GPT-4.1 | $\sim 0.95 \pm 0.05$s | 16 GB | $\sim$ $0.007 | 37.01 |

caching strategy. As shown in Table 6 This latency is $\sim 1.6\times$ OSrCIR ($\sim 0.6$s) yet slightly below CIReVL ($\sim 1.0$s), and is achieved without task-specific training. Notably, even a one-iteration setting outperforms OSrCIR* by $\sim 2.41\%$ on average at comparable cost (*i.e.,* $\sim$0.75s). Compared to textual-inversion methods, our performance surpasses them without training, but inference remains $\sim 48\times$ slower. As API calls account for 95% of inference time, faster APIs or improved tool selection could further reduce latency.

## A.7 MORE ABLATION STUDY OF TOOL-USE POLICY

In Table 7, we enforcing a rigid policy to always search or always edit yields consistent gains over disabling that capability entirely, but still falls short of the selective strategy used by the full model. We observe dataset-dependent effects: tasks with more implicit or long-tail intent (e.g., nuanced semantics) benefit more from an "always search" bias, while fine-grained, appearance-driven benchmarks favor "always edit," where a visual proxy tightens the match to subtle attributes. However, compulsory invocation introduces unnecessary calls in easy or well-specified cases, adding noise and latency without commensurate accuracy benefits. These findings reinforce our design choice: dynamic, context-aware gating with early-exit, invoking search when intent is under-specified and editing when visual grounding is needed—achieves a better accuracy–efficiency trade-off than any single fixed policy.

## A.8 EVALUATION DATASETS DETAILS

We evaluate our approach on four widely adopted CIR benchmarks: CIRR Liu et al. (2021), CIRCO Baldrati et al. (2023), FashionIQ Wu et al. (2021), and GeneCIS Vaze et al. (2023). CIRR is the first natural image dataset for CIR, but it suffers from the presence of false negatives Baldrati et al. (2023), where multiple plausible ground-truth images may exist but remain unlabeled. CIRCO addresses this limitation by providing multiple annotated ground truths per query, significantly re-

Table 7: Tool-use policy ablation on CIRCO and Fashion-IQ. Policies force the agent to *always* invoke the corresponding tool when available.

|  | CIRCO | | | Fashion-IQ | |
| --- | --- | --- | --- | --- | --- |
| Methods | k=5 | k=10 | k=25 | k=10 | k=50 |
| 1. Full model (GPT-4.1) | 27.38 | 28.96 | 31.62 | 37.94 | 59.13 |
| **Tool-use policy (appendix ablation)** | | | | | |
| 2. Always search | 27.05 | 28.02 | 31.12 | 36.78 | 58.02 |
| 3. Always edit | 26.41 | 27.95 | 30.66 | 37.13 | 58.35 |
| *Reference (from main ablation in Table 4)* | | | | | |
| w/o searching | 26.47 | 27.92 | 30.39 | 35.39 | 56.49 |
| w/o editing | 25.74 | 27.11 | 29.68 | 37.12 | 58.10 |

ducing the prevalence of false negatives. GeneCIS, constructed from MS-COCO Lin et al. (2014b) and Visual Attributes in the Wild Pham et al. (2021), supports four task variants, facilitating both object- and attribute-centric retrieval or modification around specific visual concepts. FashionIQ is focused on fine-grained, fashion-oriented retrieval driven by attribute manipulations. These datasets collectively span distinct CIR sub-tasks: CIRR and CIRCO focus on object-centric or background manipulations, GeneCIS enables compositional retrieval based on object and attribute queries, and FashionIQ emphasizes attribute-level modifications described via natural language. For evaluation, we follow the protocols established in the original benchmarks. Specifically, we report Recall@k (R@k) for CIRR, GeneCIS, and FashionIQ, and mean average precision (mAP@k) for CIRCO, to accommodate the presence of multiple ground truths. Additionally, for CIRR, we include the $\text{Recall}_{\text{Subset}}$@k metric, which measures retrieval accuracy within a restricted subset of images relevant to each query, providing a more precise assessment of compositional reasoning.

**FashionIQ Wu et al. (2021)** is a dataset of fashion-related images across three categories: Shirt, Dress, and Toptee, comprising 30,134 triplets from 77,684 images. The dataset was curated by collecting image attributes and then tasking human annotators to write captions describing highly related images based on those attributes. FashionIQ simulates realistic user interactions, as captions were generated via a chat-based visual interface to mimic online shopping queries. The dataset is divided into training (60%), validation (20%), and test (20%) splits. For zero-shot CIR, we use only the validation split, as the test set annotations are not publicly available.

**CIRR Liu et al. (2021)** contains 21,552 real-world images sourced from NLVR$^2$ Suhr et al. (2018). The dataset includes training, validation, and test splits, with the latter evaluated via a remote server. Our analysis focuses on the validation split for model selection. Unlike FashionIQ, which targets fashion-specific queries, CIRR encompasses diverse domains with complex descriptions. The dataset was built by identifying visually similar images using ResNet-152 He et al. (2016) pretrained on ImageNet Deng et al. (2009) and employing human annotators to describe differences between paired images. However, CIRR suffers from two key issues: (1) image pairs identified by ResNet often lack true visual similarity, as they were not verified by human annotators; and (2) captions are often unrealistic or ambiguous, including unnecessary details. These limitations reduce CIRR's practical relevance compared to FashionIQ. Additionally, CIRR uses a small subset retrieval task (e.g., five items) to mitigate noise, but this approach is problematic, as the target image often relates only to the text condition rather than the reference image. Previous studies Baldrati et al. (2023); Saito et al. (2023); Gu et al. (2024), have noted the prevalence of false negatives (FNs) in CIRR, complicating evaluation accuracy, as seen in other cross-modal retrieval tasks Zhu et al. (2021); Datta et al. (2008).

Notably, both FashionIQ and CIRR face challenges from FN instances. While each query has a single labeled positive, multiple valid matches may exist in the dataset. FashionIQ mitigates this by reporting Recall@K with larger K values (*e.g.,* 10 or 50), whereas CIRR employs subset retrieval. However, these approaches fail to fundamentally resolve the FN issue, particularly for CIRR's noisy annotations.

**CIRCO Baldrati et al. (2023)** builds on the COCO dataset Lin et al. (2014a), addressing the FN problem by including an average of 4.53 ground truths per query. This design enables more reliable evaluation using metrics like mAP. CIRCO contains no training split and provides validation (220 queries) and test (800 queries) splits, with the latter evaluated remotely.

**GeneCIS Vaze et al. (2023)** defines conditional retrieval tasks focusing on attributes (*e.g.,* "focus on an attribute", "change an attribute") and objects (*e.g.,* "focus on an object", "change an object"). Attribute tasks use VisualGenome Krishna et al. (2017) and VAW Pham et al. (2021), while object tasks are based on COCO Lin et al. (2014a). Each task comprises around 2,000 queries with a small gallery size (*e.g.,* 15 images, 10 for "focus on an attribute") to limit FNs. Text queries correspond to attributes or objects (*e.g.,* "color", "backpack").

Table 8: The number of images used for evaluation in each dataset.

| Dataset | Query images | Candidate images |
|---|---|---|
| CIRR (Test) | 4,148 | 2,315 |
| CIRCO (Test) | 800 | 123,403 |
| Fashion (Dress) | 2,017 | 3,817 |
| Fashion (Shirt) | 2,038 | 6,346 |
| Fashion (TopTee) | 1,961 | 5,373 |
| GeneCIS (Focus Attribute) | 2000 | 10 |
| GeneCIS (Change Attribute) | 2112 | 15 |
| GeneCIS (Focus Object) | 1960 | 15 |
| GeneCIS (Change Object) | 1960 | 15 |

A.9 EVALUATION TASKS DETAILS

**(1) Object/Attribute composition**. We evaluate the GeneCIS Vaze et al. (2023) test split and the validation split (5000 images) of COCO Lin et al. (2014a), which dataset contains images with corresponding lists of object classes and instance mask of query images. Following Pic2Word, we randomly crop one object and mask its background using its instance mask to create a query for each image. The list of object classes is used as text specification. Similarly, the GeneCIS dataset introduces four task variations, such as changing a specific attribute or object.

**(2) Object/scene manipulation by text description**. In this setup, a reference image is provided alongside a text description containing instructions for manipulating either an object or the background scene depicted in the reference image. This composition of the reference image and text description enables the retrieval of manipulated images. We evaluate the test split of CIRR Liu et al. (2021) and CIRCO Baldrati et al. (2023) using the standard evaluation protocol.

**(3) Attribute manipulation**. We employ Fashion-IQ Wu et al. (2021), which includes various modification texts related to image attributes. These attribute manipulations are given as a sentence. In evaluation, we employ the validation set, following previous works Baldrati et al. (2022); Saito et al. (2023); Baldrati et al. (2023); Tang et al. (2024).

A.10 THE USE OF LARGE LANGUAGE MODELS (LLMS)

We disclose two forms of LLM use. First, an MLLM is a core module of our method (Sec. 3): it performs selection, optional tool invocation, and refinement; model choices, prompts, decoding settings (e.g., temperature 0), seeds, and hardware are reported in Sec. 4 and detailed in the appendix (algorithms, prompt templates, and ablation protocols). Second, LLMs were used to aid writing by polishing language and formatting only; they did not originate research ideas, experimental designs, or claims, and all content was authored, verified, and curated by the authors. We provide verbatim prompt templates and configuration files in the supplementary anonymized code to support reproducibility. LLMs are not authors and are ineligible for authorship; any errors remain the responsibility of the authors. We took care to avoid plagiarism or fabrication by cross-checking all generated text against sources and our results, and by recording model names/versions for all experiments and writing assistance.

