# OpenReview forum: "Boosting Training-Free Composed Image Retrieval with Tools"
_ICLR.cc/2026/Conference — ICLR 2026 Conference Withdrawn Submission_

### Official Review · Reviewer_kgDF · 2025-10-31

**Soundness:** 3
**Presentation:** 2
**Contribution:** 2
**Rating:** 4
**Confidence:** 4

**Summary:**

In this paper, the authors propose a composed image retrieval framework that can leverage tools. The model uses a CoT prompt to dynamically select one tool out of two to improve the composed embedding. Experiments are conducted on four existing benchmarks.

**Strengths:**

1. The motivation is interesting. The authors propose a new framework for composed image retrieval that can directly leverage tools.

2. Experiments on sufficient datasets validate the performance of the proposed framework.

**Weaknesses:**

1. The presentation is not very clear. In the main paper, the authors skip many details and defer them to the appendix, which makes the main paper read like an outline.

2. It is unclear how the tool usage helps to improve the retrieval. First, whether the model can correctly select the optimal tool to be used is not clear. The authors just say they use a CoT prompt for tool selection. But I am confused whether such a prompt is the optimal. In addition, the image editing tool may change the image quality. It may produce images that are unrealistic.

3. The framework only leverages two tools, which are rather limited. In addition, both tools are existing methods or functions, making the technical contribution less impressive.

4. The inference time is concerning. The tool requires network access and API calls for each retrieval. The inference time can be significantly slower than existing methods that can directly be run locally,

**Questions:**

Please refer to the weakness above

---

### Official Review · Reviewer_VsNE · 2025-11-01

**Soundness:** 2
**Presentation:** 2
**Contribution:** 2
**Rating:** 2
**Confidence:** 3

**Summary:**

The paper focuses on training-free Zero-shot Composed Image Retrieval (ZS-CIR) methods that utilize Multimodal Large Language Models (MLLMs) to generate target captions during the retrieval phase. However, the authors identify two key limitations: (1) the domain knowledge of MLLMs is often insufficient to fully capture user intent, and (2) the generated target captions are not easily processed by standard retrievers. To address these issues, the authors propose TaCIR (Tool-augmented agent for training-free Composed Image Retrieval), which is a tool-augmented agent that jointly reasons over the reference image and manipulation text, optionally consults external tools, and instantiates the inferred edit as a visual proxy. The proposed method achieves consistent performance improvements across multiple benchmarks.

**Strengths:**

1. The proposed algorithms seem effective. It shows a large margin compared to baselines across many benchmarks.
2. The figures are illustrated well for understanding the overview.
3. There are many experiments and analyses that show the effectiveness of the proposed method.

**Weaknesses:**

1. The motivation of the paper is somewhat vague. For example, it would be helpful to include a simple experiment or analysis demonstrating that the target captions produced by the MLLM cannot be effectively processed by existing CLIP retrievers.
2. The method section also seems vague, and more details should be included.
- Regarding the domain mismatch issue, the use of knowledge acquisition and editing is reasonable. However, it is not clear how the proposed refinement procedure specifically addresses the limitation that “the target caption produced by the MLLM cannot be easily processed by the retriever.”
- The overall procedure seems computationally intensive. However, lines 466–473 claim that the method is computationally efficient compared to previous baselines. How is this possible? The “lightweight caching strategy” mentioned there is not clearly described—where in the paper is it explained, and is this strategy applicable to the baselines as well? Clarifying these points would help assess the contribution more clearly.
- The method also appears to be a combination of multiple API calls. The use of API calls (or reasoning step) in CIR might be first introduced in OsCIR, then the main contribution of the current paper would be the construction of a tool selection pool. A clearer distinction from prior methods is needed.
-For the knowledge acquisition step, it is unclear which database is used. Since the performance would heavily depend on the retrieval source, further clarification is needed. If it relies on open web search, the setting may not be fair or well-controlled.

3. The writing quality requires substantial improvement. Several sentences are unclear or grammatically incorrect, making the paper difficult to follow.
- Line 154: The phrase “pixel-level hypotheses” is unclear: please elaborate or rephrase.
- Lines 183–185: The provided example does not seem to match the figure; perhaps “manipulation text” was intended here?
- Line 186: The sentence “while Tm is a concise, executable instruction that differs from Tm” is confusing and likely contains an error.
- Line 186: The term editor is used but not clearly defined.
- The terms "implicit" and "explicit" are repeated throughout the paper, but their meanings in this context remain vague

**Questions:**

Wrote above

---

### Official Review · Reviewer_3So2 · 2025-11-04

**Soundness:** 2
**Presentation:** 2
**Contribution:** 2
**Rating:** 4
**Confidence:** 3

**Summary:**

The paper proposes a tool-augmented, training-free agent that enables multimodal large language models to use external search and image-editing tools to generate visual proxies and refined captions, thereby improving zero-shot composed image retrieval accuracy and interpretability.

**Strengths:**

- **Motivation with Tool-use for Retrieving is Good**: The motivation for developing tool-use to improve MLLM retrieving rate is good reasonable.

- **Extensive Ablations with Pipeline**: There are comprehensive ablations on different combinations and pipeline options for developing this pipeline.

- **Figures and Charts are well-designed**: The explanation with figures and charts are clear and easy to follow,

**Weaknesses:**

- **Not Clear with the Challenges**: What are the challenges for developing the pipeline here? What effort do you play with the data side? post-training side? scalable training / evaluation?  If the paper only ablates a set of options with the pipeline, it is limited and hard to justify the effectiveness and hardness for this task.

- **Hard to demonstrate the effectiveness beyond numbers**:  I can see some tables showing higher numbers across some benchmarks. However, this is not enough for such a practical use model. The paper should discuss more about more scalable evaluation and real-use cases and demonstrate the pipeline is transferable to the SOTA models like GPT-5 / Gemini / Claude models. From current performance improvement, I cannot see the effectiveness of the proposed pipeline.

**Questions:**

What is the downstream tasks and potential use cases of this pipeline? Is this pipeline scalable? How to large scale verify? Compared to the SOTA models like Gemini 2.5 pro / GPT-5 with tools, why should we follow this pipeline?

---

### Note · Authors · 2025-11-13

**Comment:**

Dear AC and Reviewers,

We sincerely appreciate the valuable feedback and constructive comments provided by the reviewers.

After careful consideration, we have decided to withdraw our manuscript.

Once again, we extend our gratitude to AC and the reviewers for your time and insightful suggestions.

Best regards,

Authors of submission 11896

**Withdrawal Confirmation:**

I have read and agree with the venue's withdrawal policy on behalf of myself and my co-authors.